# Discovering Thematically Coherent Biomedical Documents Using Contextualized Bidirectional Encoder Representations from Transformers-Based Clustering

**DOI:** 10.3390/ijerph19105893

**Published:** 2022-05-12

**Authors:** Khishigsuren Davagdorj, Ling Wang, Meijing Li, Van-Huy Pham, Keun Ho Ryu, Nipon Theera-Umpon

**Affiliations:** 1School of Electrical and Computer Engineering, Chungbuk National University, Cheongju 28644, Korea; khishigsurend@chungbuk.ac.kr; 2School of Computer Science, Northeast Electric Power University, Jilin 132013, China; smile2867ling@neepu.edu.cn; 3College of Information Engineering, Shanghai Maritime University, Shanghai 201306, China; mjli@shmtu.edu.cn; 4Data Science Laboratory, Faculty of Information Technology, Ton Duc Thang University, Ho Chi Minh City 700000, Vietnam; phamvanhuy@tdtu.edu.vn; 5Biomedical Engineering Institute, Chiang Mai University, Chiang Mai 50200, Thailand; 6Department of Electrical Engineering, Faculty of Engineering, Chiang Mai University, Chiang Mai 50200, Thailand

**Keywords:** natural language processing, pre-trained language representation model, document clustering

## Abstract

The increasing expansion of biomedical documents has increased the number of natural language textual resources related to the current applications. Meanwhile, there has been a great interest in extracting useful information from meaningful coherent groupings of textual content documents in the last decade. However, it is challenging to discover informative representations and define relevant articles from the rapidly growing biomedical literature due to the unsupervised nature of document clustering. Moreover, empirical investigations demonstrated that traditional text clustering methods produce unsatisfactory results in terms of non-contextualized vector space representations because that neglect the semantic relationship between biomedical texts. Recently, pre-trained language models have emerged as successful in a wide range of natural language processing applications. In this paper, we propose the Gaussian Mixture Model-based efficient clustering framework that incorporates substantially pre-trained (Bidirectional Encoder Representations from Transformers for Biomedical Text Mining) BioBERT domain-specific language representations to enhance the clustering accuracy. Our proposed framework consists of main three phases. First, classic text pre-processing techniques are used biomedical document data, which crawled from the PubMed repository. Second, representative vectors are extracted from a pre-trained BioBERT language model for biomedical text mining. Third, we employ the Gaussian Mixture Model as a clustering algorithm, which allows us to assign labels for each biomedical document. In order to prove the efficiency of our proposed model, we conducted a comprehensive experimental analysis utilizing several clustering algorithms while combining diverse embedding techniques. Consequently, the experimental results show that the proposed model outperforms the benchmark models by reaching performance measures of Fowlkes mallows score, silhouette coefficient, adjusted rand index, Davies-Bouldin score of 0.7817, 0.3765, 0.4478, 1.6849, respectively. We expect the outcomes of this study will assist domain specialists in comprehending thematically cohesive documents in the healthcare field.

## 1. Introduction

Active biomedical research has generated an exponentially growing, large amount of literature in modern applications. Biomedical documents are publicly accessible to the scientific community via databases, such as PubMed and PubMed Central, which contain enormous collections of research articles, online books, clinical trials, and other biomedical materials. PubMed, a search engine that primarily accesses the largest life sciences and biomedical literature library, included over 33 million articles as of April 2022 [1].

Natural language processing (NLP) in the biomedical field is successfully used in various applications, such as in extracting useful knowledge for cancer hallmarks [2], determining the information flow about the coronavirus outbreak [3], drug repurposing in diseases [4,5], healthcare recommendations based on ontology characteristics and disease [6], sentiment analysis of drug reviews [7], identification of bacterial gene expression [8], and so on. Recently, bibliometric analysis has been used to identify recent trend literature regarding different aspects of healthcare management [9]. The authors of study [9] aimed to determine factors that affect citation number, including the number of years since publication, the number of words in the title, and the number of authors of an article. Moreover, Franco et al. used a bibliometric analysis of the articles published in the last 20 years (2000–2020), exploring trends and common patterns in research of squamous cell carcinoma of the anus disease [10].

Meanwhile, various other fields have attempted to use algorithm-based text information analysis and harness its effectiveness. One of the popular methods when there is a large volume of scientific data is bibliometric analysis, which enables the identification of core patterns in the scientific community. In the field of education, a bibliometric-based systematic review was performed using publications selected from Scopus and Web of Science with related queries [11]. Results of metadata in education provided high-quality research, knowledge, and understanding of the ways in which technology can enhance education, as well as the number one source of referencing journals. Recently, an NLP system to generate geochemical and geophysical data from unstructured text was included in monitoring reports and bulletins published online [12]. The system enabled the extraction of relevant gas parameters from time series text data. In addition, it proved capable in the extraction of the time series of a set of user-defined parameters that could be later analyzed and interpreted by specialists in relation to other monitoring and geospatial data.

There is increasingly growing demand for the development of techniques to extract relevant information from huge volumes of documents associated with an input query. Therefore, document clustering, a discipline at the intersection of natural language processing and machine learning, is beneficial for a wide range of tasks, including grouping search engine results, automatic document categorization, and so on. The objective of document clustering is to discover thematically coherent documents among a vast amount of unstructured and sparse text data, and it consists of two main phases: vector representation extraction and grouping.

The huge number of unstructured and unlabeled data points are separated into discrete groups based on their comparable qualities in classic clustering algorithms [13]. However, text data contain uninformative and high dimensionality feature space that can lead to poor performance for document clustering when computing distance measures [14,15]. To overcome this problem, visualization, dimensionality reduction, or feature selection techniques are often employed beforehand [16]. Term frequency–inverse document frequency (TF–IDF) was utilized in various previous studies [17,18,19] to examine the relative frequency of words in documents. Unfortunately, TF–IDF vectors contain relatively inadequate information because dependencies and semantic relationships among concepts, as well as ordering between adjacent words, are not reflected [20].

Accordingly, document clustering models that aggregate the use of dimension reduction techniques have been investigated in order to transform the features into a reduced feature space utilizing principal component analysis (PCA) and auto-encoders (AE) techniques [21,22,23,24]. Omar [23] suggested a hybrid of statistical measures, including variance analysis, TF–IDF, and PCA, in research for selecting only and all the most distinctive characteristics that may be employed for generating document clustering tasks. As a result, their suggested model, which was fitted with a k-means clustering algorithm, was successful in reliably grouping texts within a dataset of 74 novels corpus written by 18 novelists representing various literary traditions. In a study [24], Alkhatib et al. examined multi-label text classification using two methods: semantic-based feature selection and dimensionality reduction using AE. When using the EUR-lex dataset, their experimental results demonstrated that semantic-based feature selection strategies outperformed the bag-of-word (BOW) frequency-based feature selection method using TF–IDF for feature weighting. Furthermore, they discovered that dimension reduction of original features using the AE model could still yield better results than BOW with TF–IDF.

Furthermore, many research works achieved reasonable results when using Word2Vec [25] and global vectors for word representation (GloVe) [26], which are the most well recognized word-embedding models developed by Google and Stanford, respectively. In recent times, embeddings from language models (ELMo) [27] and bidirectional encoder representations from transformers (BERT) [28] demonstrated the importance of contextualized representations derived by deeper structures such as bidirectional language models for transfer learning. These pre-trained, complex neural language models enable substantial increases in natural language processing tasks, as well as notable performance in a wide range of applications. Kong et al. [29] suggested a scientific paper recommendation system in their research. Physical Review A was used in their experiment, which had 9151 citation relationships from 7547 publications gathered between 2007 and 2009. The word-embedding techniques were performed on text information in papers of similar research interest in their suggested system. The text was represented as a vector by Doc2Vec, Struc2Vec, and DeepWalk. The authors converted structural identity to vectors in order to discover papers with comparable network topology. In another study [20], Park et al. proposed a simple and effective clustering technique called advanced document clustering. They applied GloVe, fastText, BERT, and ELMo embedding models to Squad 1.1, Yahoo Answers, REUTERS, and Fake News AMT datasets to demonstrate the efficacy of their suggested model. Moreover, they handled the high-dimensional data problem using approaches such as cosine similarity-based clustering and the mini-batch centroids update algorithm.

It is well known that using pre-training large neural language models on unlabeled text is a successful strategy for transfer learning. However, most domain-specific pre-training models are trained by starting from general-domain language models, but, in the case of specialized domains, such as the biomedical domain, document clustering differs from general-domain corpus clustering because these models are only pre-trained on general domain corpora. Thus, in this research study, we aimed to discover thematically coherent, biomedical-domain-specific, contextualized, bidirectional representations from transformers which are based on the BERT model, pre-trained on large-scale biomedical corpora [30]. Aside from identifying qualified feature representations, the challenge of clustering comparable groups based on appropriate similarity metrics is critical in document clustering applications. Evidently, clustering has been widely explored in the machine learning field in terms of feature selection, distance function, grouping techniques, and validation. Among the different grouping methods, the Gaussian mixture model (GMM) is fast and applicable to a wide variety of issues [31], including document clustering, since it employs a probabilistic assignment of data points to clusters.

In this paper, we propose a GMM-based, efficient clustering framework incorporating substantially pre-trained BioBERT domain-specific language representations to improve the clustering accuracy. Biomedical research articles crawled from the PubMed repository were utilized to develop the document clustering model for biomedical document analysis in our experiment. The proposed framework consists of three phases. In the first phase, classic text preprocessing is performed on biomedical document data. In the second phase, representative vectors for biomedical text mining are extracted from a pre-trained BioBERT language model. The third phase serves to determine the labels of biomedical documents in terms of their contextualized representative vectors using GMM. The entire modelling process is considered, including text data preparation, embedding, training, parameter adjustment, and model evaluation.

Our proposed framework is compared to several baseline models in the experimental analysis section. First, we used BioBERT, Word2Vec, TF–IDF with PCA [32], TF–IDF with AE [33], GloVe, and BioWordVec [34] embedding techniques to extract six sets of important, representative vectors from a biomedical document dataset. Second, these various representative characteristics were aggregated to provide an equal contribution to the training of three different clustering algorithms, namely GMM, k-means [35], and expectation–maximization (EM) [36]. This implies that the techniques utilized in the proposed framework could be interchanged with other comparative approaches based on the specific domain. Finally, the Fowlkes–Mallows score (FM), silhouette coefficient (SC), adjusted Rand index (ARI), and Davies–Bouldin score (DB) [37] were used to assess the effectiveness of the biomedical document clustering.

Overall, the major contributions of this paper are as follows:We developed a GMM-based efficient clustering framework that incorporates heavily pre-trained BioBERT domain-specific language representations to improve clustering accuracy for biomedical document analysis;We compared six distinct kinds of representative feature that highlight different aspects and have a significant impact on clustering effort when used in combination with different clustering techniques. The findings are useful for investigating comparable articles based on their inherent characteristics;The empirical comparison analysis demonstrates that the suggested proposed framework outperforms a variety of baseline models for biomedical-specific document analysis;The research findings are likely to contribute not only to biomedical document analysis but also to a wide range of applications in the healthcare area such as trend analysis and recommendation systems, as well as drug and gene expression identification.

The rest of this paper is structured as follows: Section 2 provides the literature review related to the topic of this research study. In Section 3, we describe the material and methods for the proposed BioBERT-based clustering framework overall experimental setup, as well as the entire procedure of analysis. Section 4 summarizes the experimental results obtained through empirical comparison analysis. Section 5 consists of a discussion of this study. Finally, Section 6 concludes with a summary of the current work and some suggestions for future investigation.

## 2. Related Work

Text mining is an essential research technique that extracts significant information from a large number of documents. Generally, the components of text mining are broadly defined as information retrieval, information processing, and information integration. For these purposes, various artificial-intelligence-based approaches are being developed. Essentially, named entity recognition (NER) is an important task that aims to automatically recognize biomedical entities, such as chemicals, diseases, and proteins, from literature. In study [38], Karatzas et al. studied a web-based Darling application for detecting disease-related, biomedical-related biomedical entity associations from disease-related PubMed literatures. Nodes in this network represented genes, proteins, chemicals, functions, tissues, diseases, environments, and phenotypes. Thereafter, Perera et al. introduced a hybrid model for food and dietary constituents named entity recognition. They also compared their proposed model with existing deep language models such as BERT, BioBERT, RoBERTa, and ELECTRA [39]. Additionally, the search tool for the retrieval of interacting genes (STRING) was used to build a protein–protein interactions network for subsequent network topology analysis [40].

Furthermore, use of conventional and modern technology in document clustering analysis is essential in biomedical area. Luo and Shah presented a biomedical text clustering framework based on disease concepts in their study [41]. To accomplish this, they extracted disease phrases and then constructed concept embeddings using neural networks. Following that, they extracted the representations using a new weighting scheme. The documents were then clustered using k-means, PCA, and t-SNE. The Trecgen collection was used in their study, which covers 4478 abstracts from the TREC 2005 Genomics Track. A clustering evaluation study found that combined word embedding and intact concept embedding-based weighting schemes outperformed TF–IDF. In study [42], Kavvadias et al. developed a web-based application for topic modelling and trend analysis of biomedical literature employing a corpus of publication titles and abstracts taken from PubMed as a source of corpora. After preprocessing the text input, the latent Dirichlet allocation (LDA) method generated the topic labels for topic modelling. Their developed application enabled the analysis of the popularity of topics over time for trend analysis and visualization.

Moreover, Muchene and Safari presented a two-stage topic modelling approach utilizing published abstract data from Kenya’s University of Nairobi in their research [43]. Firstly, they used LDA topic modelling to define per-document topic probabilities. Following that, hierarchical clustering with Hellinger distance was utilized to produce the final topic clusters, allowing the found latent topics to be reduced to clusters of homogenous topics. Their experimental findings revealed that the university’s dominating research interests included HIV and malaria research, agricultural and veterinary services research, and cross-cutting themes in the humanities and social sciences.

Karami et al. [44] attempted to identify the dominant topics of Twitter-based research, as well as to assess the temporal trend of topics and to interpret the evolution of topics. They gathered relevant publications with the word “twitter” in the title or abstract from three databases: Web of Science, EBSCO, and IEEE. Among the document analyses, LDA was used on extensive abstract texts from journals and conferences, as well as brief tweets. Finally, they discovered around 38 subjects among 18,000 research papers published between 2006 and 2019, which means that their method is useful for reviewing vast amounts of text data from any discipline and tracking changes over time. Zhang et al. [45] designed a variational neural approach for detecting biomedical event triggers that takes advantage of latent topics underlying biomedical documents. Their experimental findings demonstrated that their technique outperformed LDA, a bi-term topic model, and a document-level neural topic model on a regularly used, multi-level event extraction corpus that contained a component entity “reactive oxygen species” and a synthesis event mention.

In another piece of research [46], Liang et al. investigated the semantic representation of genes in biomedical literature to infer functional relationships using the word2vec model. The authors combined four forms of biomedical text data from biomedical articles: gene summary from RefSeq, gene reference into a function from NCBI, and gene ontology description in terms of biological process category. Their finding revealed that gene embedding might detect driver mutations, as well as improve the identification of protein complexes and functional modules. In their research [47], Boukhari and Omri presented an unsupervised biomedical document indexing approach based on approximation matching to increase the similarity between a document and a specified concept. For concept extraction, their model incorporated the vector space model with description logics. To address the non-preferred notions, a filtering step based on MeSH architecture was investigated.

Thereafter, Curiskis et al. [48] evaluated multiple approaches for document clustering analysis using three datasets from Twitter and Reddit and from online social networks. The text was extracted using Doc2Vec, TF–IDF, LDA, and Word2Vec feature representation algorithms in their comparative analysis. In addition, four clustering algorithms were examined for these representations: k-means, k-medoids, density-based spatial clustering of applications with noise, and non-negative matrix factorization. Throughout their empirical investigation, it was obvious that, when paired with k-means clustering, Doc2Vec feature representations outperformed any other combined model in three metrics, including normalized mutual information, adjusted mutual information, and adjusted Rand index. Koutsomitropoulos and Andriopoulos developed an automated medical subject headings (MeSH) model for indexing biomedical literature employing contextualized word representations in research [49]. They used biomedical literature from open-access sources, such as PubMed, EuropePMC, and ClinicalTrials, as well as hand-picked MeSH terms. The authors used two embedding methods, Doc2Vec and ELMo, to accomplish this purpose. The MeSH’s ontology representation provided machine-readable labels and determined the issue space’s dimensionality. Furthermore, they examined both deep and shallow learning methodologies. It should be highlighted that the ELMo model allowed for the construction of a multi-class classification with superior performance when compared to the Doc2Vec technique.

Another research study [50] was performed by Luo et al. The authors provided a computational framework that was developed by the following stages of utilization of a large number of clinical notes from an electronic health record. They began by extracting all possible symptom expressions using UMLS MetaMap semantic categories. Then, the pre-trained BioWordVec was used to build symptom embeddings because the seed symptom expressions in the first stage could not cover all symptom expressions stated in the clinical notes. Following that, a patient clustering approach was used to categorize the patient into groups depending on the appropriate symptom severity levels using a modified hierarchical clustering algorithm. Finally, association rule mining was used to establish the relationships between patient symptoms and risk variables. Their findings enabled physicians to detect easily hidden symptom relations and associations of patient risk factors based on clinical notes utilizing natural language modelling and machine learning techniques.

To summarize these related studies, most of the previous research studies have been conducted on TF–IDF, GloVe, Word2Vec, and Doc2Vec techniques for extracting vector representation [51]. It has also been proven that embedding techniques are definitely perform important duties for document clustering. The proposed biomedical document clustering task was realized by the GMM algorithm, which is robust and efficient in indicating the association between data instances and the cluster to which they belong. In this work, our proposed clustering model is compared and contrasted against EM and k-means clustering algorithms. For the GMM clustering algorithm, optimization is performed by applying the EM, and each independent cluster corresponds to a different Gaussian. Essentially, GMM is considered as a universal approximator of densities. On the contrary, the EM algorithm is used for various latent variable models for optimizing; it computes a lower boundary then optimizes it. Generally, the GMM is similar to the k-means model as it refines an iterative process to determine the best congestion, but k-means varies in that the centroid of each cluster is determined as the mean of all instances, whereas GMM uses the mean and variance.

## 3. Proposed BioBERT-Based Clustering Framework for Biomedical Document Analysis Materials and Methods

### 3.1. Proposed BioBERT-Based Clustering Framework for Biomedical Document Analysis

In this paper, we propose an efficient clustering framework based on the GMM, which incorporates substantially pre-trained BioBERT domain-specific language representations for biomedical document analysis, in order to improve the clustering effort. The proposed framework, as illustrated in Figure 1, consists of three sections: data preprocessing, representative feature extraction, and document clustering.

#### 3.1.1. Data Preprocessing

Real-world datasets include essential information, but they are not in a format appropriate for the required data analysis technique. Thus, they need to be cleaned to strengthen the clustering effort. In this study, we eliminate documents that utilize any languages other than English in order to tackle the problem of multiple languages. The text-cleaning process deletes stopwords, unique features, single numbers, and noises from the English text. This cleaning process is carried out by employing regular expressions to execute automated searches in the text for certain specific patterns in order to eliminate them.

Following that, common tasks based on tokenization, stemming, and lemmatization are completed. Tokenization is the process of splitting a text document into smaller units, such as individual words. Both stemming and lemmatization produce the root form of the inflected words. Stemming simply eliminates or stems the last few characters of a word, which frequently results in wrong interpretations and spelling. Lemmatization typically refers to performing things correctly by using a vocabulary and morphological analysis of words, with the intention of eliminating only inflectional ends and returning the base or dictionary form of a word, which is referred to as the lemma.

#### 3.1.2. Bidirectional Encoder Representations from Transformers for Biomedical Text Mining

BioBERT is a language representation model that has been pre-trained for the biomedical domain. BioBERT is completely initialized with weights from the BERT model, which leverages the attention mechanism in a transformer-based architecture. On a variety of natural language processing tasks, the BERT model has shown remarkably effective and empirically impressive outcomes. The BERT model is primarily composed of two steps: pre-training and fine tuning. For pre-training BERT, the biggest corpus of BooksCorpus (0.8 billion words) and English Wikipedia (2.5 billion words) were employed. However, because it was pre-trained in the broad domain, directly applying BERT to biomedical document analysis is limited in terms of extracting relevant information and understanding biomedical texts.

Fortunately, BioBERT has been pre-trained on a large number of biomedical domain corpora derived from PubMed abstracts and PMC full-text articles, employing the same architecture as BERT. In addition, the batch size, learning rate, and other parameters for pre-training BioBERT were the same as for pre-training BERT. In a variety of biomedical text mining tasks, BioBERT exceeds several state-of-the-art models. Therefore, vector representations generated from the BioBERT model are utilized in our biomedical document analysis study.

Table 1 presents detailed information on pre-trained corpora for BioBERT. Moreover, different text corpora combinations were investigated in BioBERT, including BERT (Wiki + Books) + PubMed, BERT (Wiki + Books) + PMC, and BERT (Wiki + Books) + PubMed + PMC. We utilize the accessible version of pre-trained weights for BioBERT—Base v1.0 (BERT + PubMed 200K + PMC 270K) from https://github.com/dmis-lab/biobert (accessed on 20 March 2022) in our experiment.

The overall architecture of BioBERT is depicted in Figure 2. In this architecture, the input representation for each given token is produced by aggregating the associated token, segmentation, and position embeddings. The input text is separated into sections by the special symbols [CLS] and [SEP]. [CLS] is positioned in front of each input text and is used in classification layers. [SEP] is a special separator for joining two sentences that indicates whether a token belongs to sentence a or sentence b.

BioBERT utilizes bidirectional transformers, as opposed to earlier standard language models such as the OpenAI generative pre-trained transformer (GPT) and ELMo. In the case of OpenAI GPT, a left-to-right transformer is utilized, but, in the ELMo model, a concatenation of separately trained left-to-right and right-to-left LSTMs is used to create features for downstream tasks. Only BioBERT representations are jointly conditioned on both left and right contexts within layers, as opposed to OpenAI GPTs and ELMo. Due to the difficulty of language modelling in which future words cannot be seen, BioBERT employs masked language models, which implies that part of the input tokens is masked at random throughout the training process, and these masked tokens are then predicted. Furthermore, the task of predicting the next word is completed in order to enable bidirectional representations; therefore, BioBERT has shown that it can overcome the limitation of a unidirectional language model.

#### 3.1.3. Gaussian Mixture Model

The GMM is a probability density function that is expressed as a weighted sum of several Gaussian densities. GMM is used to iteratively estimate a collection of parameters until the desired convergence value is reached. GMM makes use of a fixed number of Gaussian distributions to determine the number of clusters in the dataset. GMM is defined as the weighted sum of M Gaussian densities, as provided by the equation:(1)$$p\left(x|\lambda \right)=\overset{M}{\underset{i=1}{\sum }}w_{i}g\left(x|\mu_{i},{\sum^{\;}}_{i}\right)$$
where x is a data vector, $w_{i},\;i=1,\;2,\;3\;\ldots ,\;M$ are the mixture weights, and $g\left(x|\mu_{i},{\sum^{\;}}_{i}\right),\;i=1,\;2,\;3\;\ldots ,\;M$ are the Gaussian densities. Each density is a d-variate Gaussian function of the following form:(2)$$g\left(x|\mu_{i},\Sigma_{i}\right)=\frac{1}{{\left(2\pi \right)}^{\frac{D}{2}}{\left|\Sigma_{i}\right|}^{\frac{1}{2}}}exp\left\{-\frac{1}{2}{\left(x-\mu_{i}\right)}^{\prime }\sum_{i}^{-1}\left(x-\mu_{i}\right)\right\}$$
where $\mu_{i}$ is the mean vector defining the center, and ${\sum^{\;}}_{i}$ is the covariance defining its width. It is equivalent to the dimensions of an ellipsoid in a multivariate scenario. The mixture weights satisfy the constraints that $\overset{M}{\underset{i=1}{\sum }}w_{i}=1$.

Each Gaussian in the GMM is made up of parameters, such as mixture weights ($w_{i}$), mean vectors ($\mu_{i}$), and covariance matrices (${\sum^{\;}}_{i}$), which are denoted by the following notation:$$\lambda =\left\{w_{i},\mu_{i},\;{\sum^{\;}}_{i}\right\}$$

Estimating the appropriate parameters configuration of a GMM for training vectors is a critical challenge. In this work, maximum likelihood estimation, the most common and well-established approach, is used to discover model parameters in order to maximize the likelihood of the GMM. In addition, the expectation–maximization approach is used to estimate the model parameters.

### 3.2. Experimental Setup

#### 3.2.1. Experimental Environment

All experiments were carried out on a computer running Microsoft Windows 10 and equipped with an Intel Core i5-6600K CPU 3.50 GHz, an NVIDIA GeForce RTX 2060, and 32 GB of RAM. Python 3.8 (Python 3 Reference Manual, Scotts Valley, CA, USA: CreateSpace) was used as the programming language. To develop the comparison experiments, general machine learning and NLP tools from Gensim, NTLK, Tensor-flow, Scikit-learn, and other libraries were deployed.

#### 3.2.2. Baseline Models


**Embedding Techniques**


**Word2Vec:** Word2Vec was developed by Mikolov at Google, and it is a standard two-layer neural network trained to generate a dense vector with a given dimension for each word.

The Word2Vec model employs the skip gram and continuous bag of words (CBOW), as illustrated in Figure 3. Given a word, the skip-gram model predicts its context. The CBOW model is the inverse of the skip-gram model. Word2Vec, which consists of 300-dimensional vectors for 3 million words, is used as a baseline model in our experimental analysis.

**GloVe:** GloVe is an unsupervised learning algorithm that generates vector representations for words. The GloVe model works similarly to the Word2Vec model, with the primary distinction being that GloVe trains on global co-occurrence counts rather than discrete local context windows as in Word2Vec. The GloVe contains 840 billion 300-dimensional word vectors for text representation.

**TF–IDF:** TF–IDF statistics estimate the relevance of a term to a document in a corpus. This method is accomplished by multiplying two metrics. Initially, it quantifies a word identified in a document. Then, for each word in the document, it assigns weight as follows:(3)TF−IDF (t,d,D)=tf (t, d)×idf (t,D)
where tf (t, d) is the number of occurrences of each word in each document. In this case,
(4)$$idf\;\left(t,D\right)=\log\left(\frac{\left|D\right|}{1+\left|d\in D\;:t\in d\right|}\right)$$

The concept that the specificity of a term may be determined as an inverse function of the number of documents in which it occurs is defined by idf.

PCA and AE are dimension reduction techniques used for TF–IDF vectors to avoid dimensionality problems before clustering, as shown in Figure 4 and Figure 5. PCA is commonly used in exploratory data analysis, which involves a dataset including observations on p numerical variables and n data values. These data values specify p n-dimensional vectors of $x_{1},\;x_{2},\;x_{3}\ldots \;x_{p}$, which is similar to n×p data matrix M, the $j^{th}$ column of which represents the vector $x_{j}$ of observations on the $j^{th}$ variable. PCA is used to transform data representations by geometrically projecting them into a lower number of dimensions while seeking a linear combination of the columns of matrix M with maximum variance. The main concept behind this technique is to generate principal components in order to minimize the gap between data and principal components, whereas it maximizes the variance of the projected data. PCA obtains the orthogonal subspace because eigenvectors (projected data) obtained from eigendecomposition are applied to a covariance matrix.

AE is a relatively new approach, an unsupervised artificial neural network that determines data representation in a lower dimension. The architecture of AE comprises an encoder and a decoder, where the encoder compresses the data to a lower dimension and the decoder takes the lower-dimensional data and reconstructs the input dataset. It has a single input and a single output layer in general, with neurons in the output layer associated with each input. Thus, the number of neurons in the output layer equals the number of neurons in the input layer. The main learning process of AE is to compress input data into a reduced number of dimensional spaces, which is referred to as a latent space. Following that, it decodes the compressed input data and generates an output. Finally, its goal is to reconstruct its inputs rather than predict a target value Y given an input X while minimizing the difference between input and output.

**BioWordVec:** BioWordVec is an open collection of biological word vectors. First, it integrates sub-word information from biomedical texts with biomedical subject headings and a biomedical-controlled vocabulary (MeSH). The fastText sub-word embedding model is then used to learn the distributed word embedding from text and MeSH term sequences. This sub-word embedding improves the quality and semantics of biomedical word representations, which is valuable for biomedical natural language processing applications.

The models outlined above are employed in our study to handle biomedical document data in various ways, generating various types of vector representation. Furthermore, as previously stated, various embedding models differ in unit and level. Table 2 shows the unit utilized by each model, as well as the level at which they operate.


**Clustering Techniques**


**K-means:** The k-means clustering algorithm is extensively used for splitting a given dataset into K groups. To do this, K cluster centroids are randomly initialized, which is a user-specified parameter. Then, as a cluster prototype, a collection of points is allocated to the cluster centroids based on their distance. Following that, the centroid of each cluster is updated depending on the points allocated to the cluster. This update procedure is performed until a specified convergence criterion is satisfied.

**EM:** The EM technique is an iterative approach for estimating parameters in probabilistic models using maximum likelihood. The expectation (E) and maximization (M) steps are alternated in EM. In the expectation step, it guesses the required parameters based on the observed data points. In the maximization step, it computes parameters that maximize the expected log likelihood determined in the expectation step. In other words, the values of latent variables are estimated in the E step, the model is optimized in the M step, and the process is repeated until it converges to a local optimum.

#### 3.2.3. Evaluation Metrics

**FM:** The FM score is an assessment metric that is used to determine clustering similarity. The FM score is defined as the geometric mean of a clustering’s precision and recall values as follows:(5)$$FM=\sqrt{prec\cdot recall}=\frac{TP}{\sqrt{\left(TP+FN\right)\left(TP+FP\right)}}$$
where *TP* denotes the number of points in the same class and cluster, *FP* defines the number of points in a class but not in a cluster, and *FN* denotes the number of points in different classes and clusters. Precision is the fraction of a cluster that contains points of a specific class. While the recall is defined as the number of relevant points retrieved by a search divided by the total number of relevant points.

**SC:** The SC is a popular method that combines cluster cohesion and separation. It computes the average distance between each point in the cluster. Furthermore, when points are not contained in the cluster, it computes the average distance between the point and all points in the nearest cluster. The following formula is used to determine such a value with regard to all clusters:(6)$$S_{i}=\frac{\mu_{out}^{min}\left(x_{i}\right)-\mu_{in}\left(x_{i}\right)}{max\left\{\mu_{out}^{min}\left(x_{i}\right),\mu_{in}\left(x_{i}\right)\right\}}$$
where $\mu_{in}\left(x_{i}\right)$ is the mean distance from $x_{i}$ to points in its particular cluster, and $\mu_{out}^{min}\left(x_{i}\right)$ is the mean of the distances from $x_{i}$ to points in the nearest cluster. The $S_{i}$ value of a point remains between [−1,+1] intervals. A value close to +1 indicates a well-defined grouping.

**ARI:** The ARI computes the similarity between two clusterings by estimating all points and counting points that belong in the same or different clusters in the predicted and true clusterings, as shown below:(7)$$ARI=\frac{Index-Expected\;Index}{Max\;Index-Expected\;Index\;}$$

The similarity score ranges from −1.0 to 1.0. When the ARI is between 0.0 and 1.0, it indicates that the clustering is good.

**DB:** The DB index measures how compact the clusters are in comparison to the distance between the clusters’ means. The index is then computed in the following manner:

Let $\mu_{i}$ indicate the cluster means, which is known as:(8)$$\mu_{i}=\frac{1}{n_{i}}\underset{x_{j}\in C_{i}}{\sum }x_{j}\;$$
(9)$$\sigma_{\mu_{i}}=\sqrt{\frac{\sum_{x_{j}\in C_{i}}\delta {\left(x_{j,}\mu_{i}\right)}^{2}}{n_{i}}}=\sqrt{var\left(C_{i}\right)\;}$$
(10)$$DB=\frac{1}{k}\overset{k}{\underset{i=1}{\sum }}\underset{j\neq i}{\max}\left\{DB_{ij}\right\}$$
where $\sigma_{\mu i}$ denotes the dispersion of the points around the cluster means, and $var\left(C_{i}\right)$ is total variance. The minimum score is zero, and lower values indicate better grouping.

## 4. Experimental Result and Analysis

In this section, we summarize the overall comparison results that were attained over our proposed efficient deep clustering framework and various computable baseline models for biomedical document analysis.

### 4.1. Text Data Preprocessing

In this study, we collected 5750 research publications from the PubMed repository that were possibly related to the biomedical sector between 2017 and 2020. First, we performed common text preprocessing techniques, including removing non-ASCII characters, punctuation, numbers, articles, and stopwords after transforming upper case to lower case. During literature text analysis, we expanded the NLTK stopword list to include terms such as ‘recent’, ‘abstracttext’, ‘stringElement’, ‘background’, ‘backgrounds’, ‘results’, ‘conclusions’, ‘string’, ‘materials’, ‘text’, ‘element’, ‘model’, ‘year’, ‘method’, ‘methods’, ‘inform’, ‘study’, ‘label’, ‘nlmcategory’, ‘copyright’, and so on. We eliminated these non-medical common words because they appeared often in the literature and might have served as noise in the clustering procedure. Then, tokenization, stemming, and lemmatization techniques were used to provide tokens for the model. As a result, the first step worked to clean up the text data and prepare it for the later phases of vector representation and modelling.

### 4.2. Comparison Results of Clustering Models

The proposed GMM-based biomedical document clustering framework incorporates the use of the pre-trained BioBERT language model. The BioBERT model in this experiment comprised of 12 encoder layers known as transformer blocks and 12 self-attention heads per transformer layer, as well as an input of dimensions 768 by taking no more than 512 tokens in an input sequence. A batch size of 64 and a learning rate of 3 × 10^−5^ were chosen for fine tuning. GMM was used to compute the class number of the clusters, with parameters, such as covariance type being ‘complete’, which implies that each component has its own general covariance matrix, convergence threshold being 0.001, and number of EM iterations being 100. The number of initializations (n_init) parameter shows how many times the GMM is initialized. We set n_init to 10 times; it can decrease the chance of converging on insufficient clusters, as well as retain the best results. The elbow approach essentially involves adjusting the number of clusters provided as input between 2 and 15 when training a GMM-based BioBERT model.

Table 3 shows the evaluation results of the GMM clustering model aggregated with different representations of BioBERT, Word2Vec, TF–IDF with PCA, TF–IDF with AE, GloVe, and BioWordVec techniques on biomedical documents. We evaluated the model using four evaluation metrics: FM, SC, ARI, and DB. As demonstrated in Table 3, the contextualized representations of the BioBERT-based model performed the best in terms of clustering, with an FM score of 0.7817, ARI of 0.4478, and DB of 1.6849.

Meanwhile, the BioWordVec-representations-based GMM model achieved the greatest SC of 0.3854. Furthermore, the BioWordVec-based model achieved the second best results, with an FM score of 0.7621, ARI of 0.4095, and DB of 1.7309, because it integrated sub-word information from biological literature with biomedical subject headings. In the cases of the Word2Vec and GloVe models, they achieved inferior results when compared to contextualized vector representative models of BioBERT and BioWordVec. In contrast, the FM, SC, ARI, and DB scores for the TF–IDF using the PCA-based model were 0.5308, 0.0969, 0.0863, and 4.5658, respectively. When employing AE as dimension reduction, TF–IDF vectors performed marginally better than PCA-based models, with an FM score of 0.5659, SC of 0.0493, ARI of 0.0751, and DB of 3.785, respectively. These evaluation findings of the GMM-based contextualized BioBERT clustering model clearly demonstrate the superiority of several embedding approaches in terms of clustering performance.

Nonetheless, as indicated in Table 4 and Table 5, these varied representative characteristics were combined to offer an equivalent contribution to the training of different clustering algorithms of k-means and EM.

The maximum number of iterations within each run for the k-means clustering algorithm was 300, and the relative tolerance was 1 × 10^−4^. Additionally, the k-means algorithm initialized the centroid 10 times and returned the most converging values as the best result. According to the comparison findings of the k-means algorithm, the best representative vectors were generated by BioBERT, which achieved an FM score, SC, ARI, and DB of 0.7712, 0.3041, 0.4369, and 1.8507, respectively. Thereafter, BioWordVec vector representations seemed to have the second highest scores, with a FM score of 0.7283, SC of 0.2624, ARI of 0.4294, and DB of 1.9204. Following that, GloVe achieved an FM score of 0.5929, SC of 0.2658, ARI of 0.2904, and DB of 2.8612, which were slightly better than the Word2Vec model. The TF–IDF representative vectors-based k-means algorithm, like the GMM-based models, differentiated the lowest scores even when vector dimensions were reduced.

As shown in Table 5, while evaluating the performance of high-performing models using the EM clustering approach, BioBERT exhibited superior vector representation versus others. Contextualized representations from BioBERT with the EM model achieved the best FM score of 0.6798 and ARI of 0.3258. In terms of the data, BioWordVec demonstrated the best biomedical document clustering capabilities. Therefore, it is clear that the employment of a contextualized vector representative model is required to comprehend domain-specific document clustering. Furthermore, the BioWordVec and BioBERT language models yielded better EM clustering results. The GloVe had the best FM, SC, ARI, and DB scores among the non-contextualized vector representations at 0.5573, 0.2355, 0.2216, and 3.8381, respectively.

**Table 5 ijerph-19-05893-t005:** Evaluation results of expectation–maximization clustering for biomedical documents.

Representations	Fowlkes–Mallows Score	Silhouette Coefficient	Adjusted Rand Index	Davies–Bouldin Score
BioBERT	0.6798	0.2762	0.3258	2.2121
Word2Vec	0.5339	0.2112	0.0875	4.8135
Glove	0.5573	0.2355	0.2216	3.8381
TF–IDF with PCA	0.4545	0.0752	0.0626	5.7447
TF–IDF with AE	0.5102	0.1118	0.0684	4.8371
BioWordVec	0.6356	0.2995	0.3055	1.9482

Following that, Word2Vec, TF–IDF with AE, and TF–IDF with PCA models were sorted in descending order of their EM clustering capabilities. Figure 6 depicts a chart of clustering models of biomedical documents for each of the evaluation metrics. The X-axis in these figures shows the adopted methods, while the Y-axis represents the utilized evaluation metrics. These diagrams show how the clustering results altered based on the vector representations and grouping algorithms. When the results of these multiple models were carefully examined, contextualized document representations from the pre-trained BioBERT outperformed feature-based Word2Vec, TF–IDF with PCA, TF–IDF with AE, and GloVe representations in terms of overall evaluation performance. Furthermore, not only BioBERT but also BioWordVec exceeded the competition, achieving computable results across all vector extraction techniques.

In terms of FM score, it is obvious that GMM-based clustering results outperformed k-means and EM clustering algorithms when aggregated with six distinct representations. Even when integrated with three clustering techniques, TF–IDF with PCA-based vector representations recorded the lowest FM score when compared to other vector representations. Following that, TF–IDF with AE, Word2Vec, and GloVe models showed somewhat higher FM scores. As can be noticed, representative vectors are critical for improving clustering accuracy.

For SC analysis, the GMM clustering model integrated with BioWordVec and BioBERT contextualized embeddings identified as a promising combination of clustering models for biomedical documents. On the other hand, TF–IDF vectors achieved the lowest SC. In addition, when compared to the PCA model, the TF–IDF with AE-based vectors with k-means and EM had better SCs of 0.0867 and 0.1118, respectively. Remarkably, the TF–IDF with AE vectors combined with the GMM model behaved 0.0476 times worse than the TF–IDF with PCA-based GMM. With regard to the ARI, our proposed GMM demonstrated the best biomedical document clustering ability when equipped with BioBERT. Among the compared clustering algorithms, the k-means clustering algorithm provided computable results with GMM. Most TF–IDF vector-based models remained the worst, but the TF–IDF with AE-based k-means model performed well, with an ARI of 0.2758, which was a significantly higher score than other TF–IDF vector-based representations. Therefore, when compared to non-contextualized vector representation models, GloVe-based models were recognized as an acceptable model in terms of ARI.

According to the DB score, the lowest values indicate better clustering effort. Thus, our proposed GMM-based clustering framework incorporated with BioBERT achieved a remarkably significant score of 1.6849. It should be highlighted that the BioBERT-based k-means clustering model produced computable findings and was ranked second in this experimental investigation of biomedical literature clustering. Considering the clustering techniques utilized in our investigation, EM achieved the worst DB scores.

Among the FM, SC, ARI, and DB evaluation measures, it was recognized that AE-based TF–IDF vectors commonly achieved marginally better results than PCA-based TF–IDF models. Moreover, when combined with the EM clustering technique, PCA-based TF–IDF obtained the lowest FM score of 0.4545, SC of 0.0623, ARI of 0.0626, and DB of 5.7447. Because tokens in TF–IDF are not examined sequentially, dependencies and connections between tokens cannot be reflected in TF–IDF vectors, resulting in less informative and ineffective representations. Therefore, the GMM clustering model integrated with pre-trained contextualized representations from the BioBERT model was rated as the best, big deep clustering model for biomedical document analysis.

Moreover, as we focused more on selecting representative vectors in this experimental analysis, relatively few parameters were used in each clustering algorithm. Particularly, further research is needed to develop the clustering algorithm in order to reduce the standard errors with capable parameter tuning and selecting a more fitting algorithm. As shown in Figure 7, we use the “WordCloud” Python module to create visual representations for the nine clusters of biomedical documents. A word cloud is a sophisticated graphical representation that is used to illustrate the words that appear the most frequently in each document. Otherwise, the most frequently occurring terms in a document cluster appear larger.

## 5. Discussion

Earlier embedding techniques, such as TF–IDF or LDA, focus on independent representations. However, polysemy is one of the most difficult issues for conventional word embeddings, which indicates that a word can have multiple meanings depending on the context [52,53]. Nevertheless, the majority of current research studies focused on learning context-dependent representations. Substantial improvements in natural language processing were made possible by the development of transformers [54], such as the transformer-based architectures of the BERT model. BERT is a particularly unique pre-training method because it is built on a masked language model with bidirectional transformers. Furthermore, ELMo and BERT demonstrated the best results in a variety of natural language processing applications. The information in the BERT model comes from bidirectional representations rather than unidirectional representations, which is crucial for the purpose of the word representation.

However, these models are unable to extract the most effective representative vectors in the biomedical domain because they are pre-trained using broad-domain corpora. Fortunately, BioBERT, a domain-specific language representation model pre-trained on large-scale biomedical corpora, was introduced by Lee et al. [30]. In general, they trained the pre-trained language model BERT on biomedical corpora collected from PubMed abstracts (PubMed) and PubMed Central full-text articles (PMC). According to their findings, BioBERT outperformed state-of-the-art models in biomedical text mining tasks such as biomedical named entity recognition, biomedical relation extraction, and biomedical question answering. Due to the BioBERT model’s effectiveness, we utilized it in our research study for biomedical document analysis.

For the research area of biomedical document analysis, our main contribution is our proposed GMM-based efficient clustering framework that incorporates heavily pre-trained contextualized bidirectional encoder representations from the transformers-based clustering model. In the experimental results and analysis section, it is also well demonstrated that the best model was distinguished by our proposed model in terms of FM, SC, ARI, and DB scores, which reached 0.7817, 0.3765, 0.4478, and 1.6849, respectively. Another difference from related studies is that we compared six distinct kinds of representative vector and also determined their significant impact on several clustering algorithms. The evidence from the results suggests that TF–IDF vector-based models give the worst results, even if combined with dimension reduction techniques of the PCA and AE, as shown in Figure 6. As seen from a careful look at the results of TF–IDF with PCA and AE models, AE was highly effective for high-dimensional text data. Nonetheless, domain-specified embeddings performed more significant scores for discovering thematically coherent biomedical documents. We expect that the aggregated different models and their findings give more research motivation to domain experts.

## 6. Conclusions and Future Work

A massive volume of biomedical literature has been growing exponentially. Meanwhile, due to the unsupervised nature of the large-scale biomedical literature database, it is difficult to discover relevant papers. To enhance clustering accuracy, we proposed a GMM-based efficient clustering framework incorporating substantially pre-trained BioBERT domain-specific language representations. This framework’s procedure is divided into three phases: (I) collection and preprocessing of biomedical document data; (II) generation of representative vectors from a pre-trained BioBERT language model; and (III) tuning and utilization of GMM to construct the clustering model. Furthermore, we compared several baselines to our proposed framework to demonstrate the effectiveness of each phase. As a consequence, the proposed framework achieved a notably superior clustering performance. It should be noted that empirical comparison results showed that contextualized representative vectors extracted from a heavily pre-trained BioBERT language model reached better capable clustering efforts in biomedical document analysis. These findings will be useful for investigating comparable articles based on their inherent characteristics and can also contribute to a wide range of applications in the healthcare area. A further extended analysis is expected to develop real-time biomedical textual information collection architecture in a big data environment.

## Figures and Tables

**Figure 1 ijerph-19-05893-f001:**
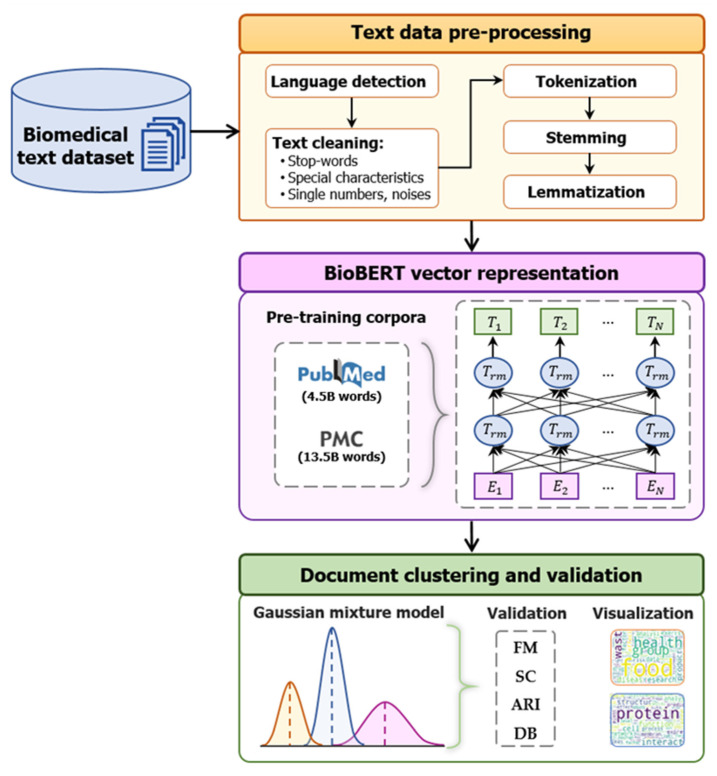
Contextualized bidirectional encoder representations from transformers-based clustering framework for biomedical documents.

**Figure 2 ijerph-19-05893-f002:**
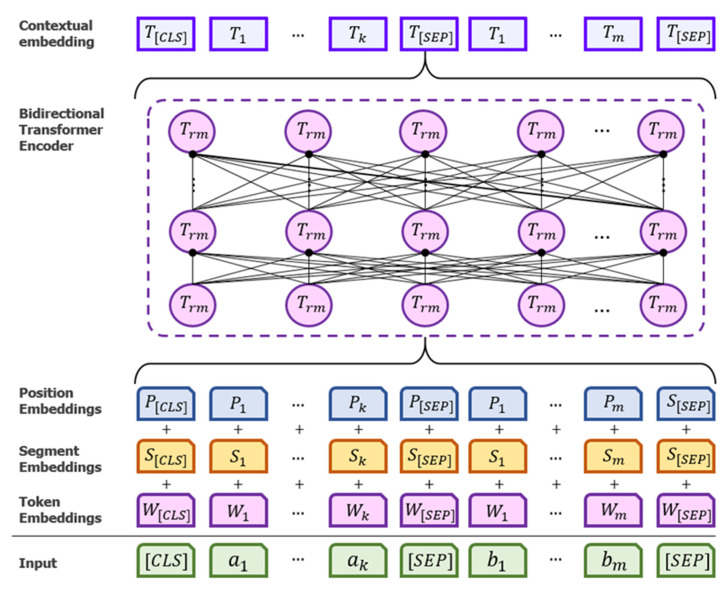
General architecture of BioBERT model.

**Figure 3 ijerph-19-05893-f003:**
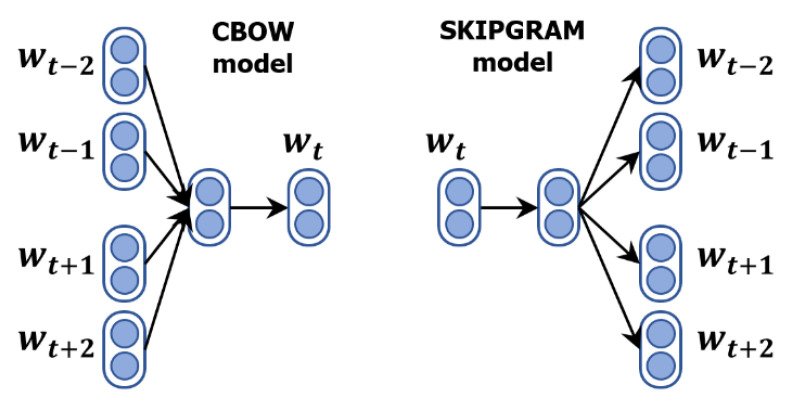
The architecture of the Word2Vec models: CBOW and skip gram.

**Figure 4 ijerph-19-05893-f004:**
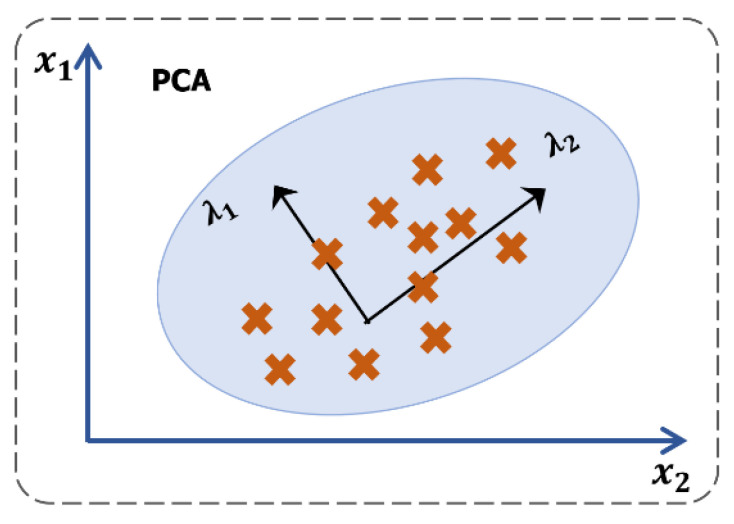
The architecture of principal component analysis.

**Figure 5 ijerph-19-05893-f005:**
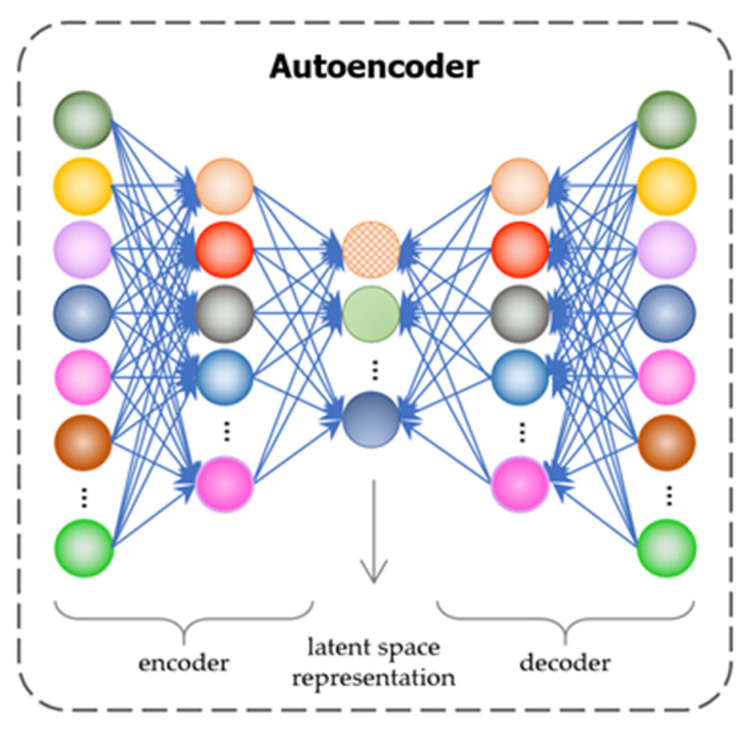
The architecture of the auto-encoder model.

**Figure 6 ijerph-19-05893-f006:**
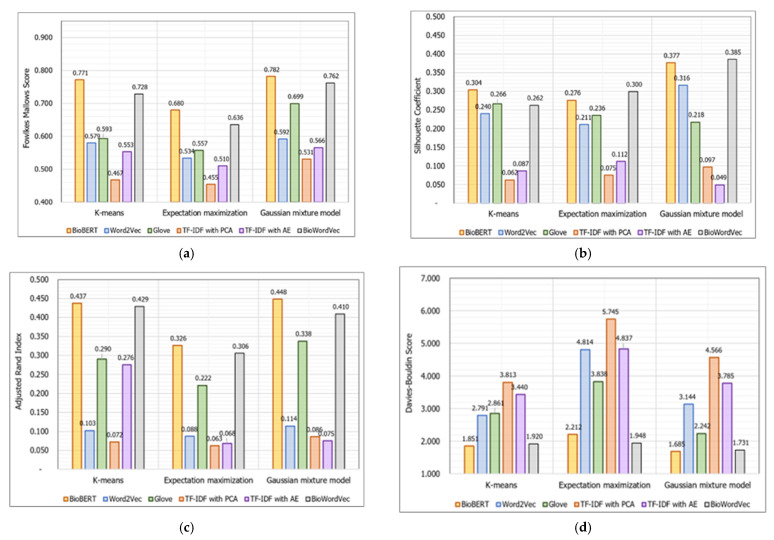
Comparison charts of the biomedical document clustering models based on different evaluation metrics: (**a**) Fowlkes–Mallows score, (**b**) silhouette coefficient, (**c**) adjusted Rand index, (**d**) Davies–Bouldin score.

**Figure 7 ijerph-19-05893-f007:**
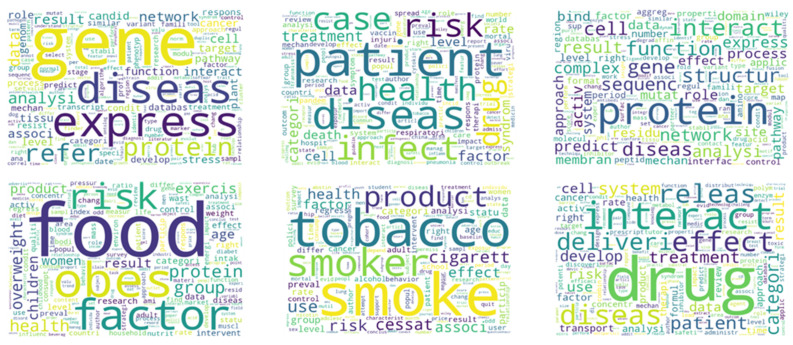
Word cloud representations for the biomedical documents clusters.

**Table 1 ijerph-19-05893-t001:** List of pre-trained corpora for BioBERT.

Corpus	Number of Words (by Billion)	Domain
BooksCorpus	2.5	General
English Wikipedia	0.8	General
PubMed abstracts	4.5	Biomedical
PMC full-text articles	13.5	Biomedical

**Table 2 ijerph-19-05893-t002:** The vector representation models and their characteristics.

Name	Unit	Level
BioBERT	Contextual string embedding	Sentences
Word2Vec	Words	Local context
TF–IDF	Words	Corpus
GloVe	Words	Corpus
BioWordVec	Contextual sub-word	Local context

**Table 3 ijerph-19-05893-t003:** Evaluation results of Gaussian-mixture-model-based clustering for biomedical documents.

Representations	Fowlkes–Mallows Score	Silhouette Coefficient	Adjusted Rand Index	Davies–Bouldin Score
BioBERT	0.7817	0.3765	0.4478	1.6849
Word2Vec	0.5919	0.3162	0.1143	3.1435
GloVe	0.6994	0.2175	0.3375	2.2419
TF–IDF with PCA	0.5308	0.0969	0.0863	4.5658
TF–IDF with AE	0.5659	0.0493	0.0751	3.7854
BioWordVec	0.7621	0.3854	0.4095	1.7309

**Table 4 ijerph-19-05893-t004:** Evaluation results of k-means clustering for biomedical documents.

Representations	Fowlkes–Mallows Score	Silhouette Coefficient	Adjusted Rand Index	Davies–Bouldin Score
BioBERT	0.7712	0.3041	0.4369	1.8507
Word2Vec	0.5794	0.2395	0.1025	2.7911
GloVe	0.5929	0.2658	0.2904	2.8612
TF–IDF with PCA	0.4672	0.0623	0.0719	3.8127
TF–IDF with AE	0.5531	0.0867	0.2758	3.4395
BioWordVec	0.7283	0.2624	0.4294	1.9204

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
