# Peer review of "Discovering Thematically Coherent Biomedical Documents Using Contextualized Bidirectional Encoder Representations from Transformers-Based Clustering"

_ijerph, 2022, doi:10.3390/ijerph19105893_

Round 1

Reviewer 1 Report

  1. Please indicate the originality of this study in Abstract.
  2. The contribution and research question of the article need to be more clearly articulated. Currently, this paper lacks a strong breakdown of what exactly is being studied, why, and what the major contribution is.

  3. Some of the important statements lack relevant references.
  4. I recommend the authors improve and restructure the discussion and conclusion parts.
  5. The conclusions are simplistic and lack in-depth analysis

Author Response

Dear Reviewer, MDPI International Journal of Environmental Research and Public Health
Thank you for giving valuable comments.
According to your comments, we revised and uploaded (a) our point-by-point response to the comments, (b) an updated manuscript with yellow highlighting indicating changes, and (c) a clean updated manuscript without highlights.
Please, find the attached response [answer sheet (a)] for your comments.

I hope our manuscript is suitable for publication.
Thank you for your comments.  

Best regards,
Keun Ho Ryu

Reviewer 2 Report

The article consists of total 20 pages, including 7 figures, 4 tables, 10 mathematical formulas and the list of total 46 literature references. The manuscript presents as original material-based article that describes techniques of mass-scale analyzing of the contents of electronic text documents and then finding and grouping documents according to the topics they have in common. It is one of the current vital scientific demands to find new ways of identifying the most question-relevant data in available vast data collections. As such, the topic of the manuscript fits into the scope of works published in the Journal. The manuscript is interesting and written in good quality, communicative English but following the presented ideas still may be demanding for Readers who are not specializing in the respective quite narrow field. The structure of the manuscript may require adjustment for the coherence with the widely accepted text structure: introduction-materials and methods-results-discussion-conclusion; the current contents require just adding the respective headers but it will add a lot to the clarity of organization of the text.

The title of the document might hint more overtly at the overall contents and general topic of the article e.c. “Discovering thematically coherent biomedical documents using contextualized bidirectional encoder representations from transformers based clustering” as the current version of the title does not bring any associations in Readers who are not narrowly specialized in data mining while the contents of the article may be interesting for a much wider audience.

The Abstract is not structured but it provides a relevant core outline of the contents of the main text.

The Introduction (with Related Work) section provides the Readers with disciplined and comprehensive review of the vital background information on history, development and current status of the field the presented study concerns, enriched with relevant clarifying figure.

There is no Material and Methods section - but the current sections No 3 and 4 with their sub-sections shall be organized into one main section by adding the heading Materials and Methods and treating the current main sections as its sub-sections. The contents document in high detail the methodology applied by the Authors.

The Results section is presented in enough detail and consistent with the presented study methodology.

There is no separate Discussion section in the current version of the text and the Authors shall consider adding one, especially as the text contains so through and detailed introductory review of the current status of the field and there is room for at least some comparison of the Authors’ achieved own results against other previously published works in the field.

The Conclusion section is clear enough and consistent with the presented results.

The Authors may consider adding to their article introduction same broader and practical background aspects concerning the data processing science-related field as referred to e.c. in the following sources:

  • various fields attempts to use algorithm-based structured and unstructured text information analysis and their effectiveness, like it is e.c. here: https://doi.org/10.3390/app12073503 https://doi.org/10.3390/math10060993 https://doi.org/10.3390/make4010012 and https://doi.org/10.3390/educsci12030210
  • attempts to use algorithm-based medical text information analysis and their effects, like it is e.c. here: https://doi.org/10.3390/cancers14071697 https://doi.org/10.3390/biomedicines10030724 https://doi.org/10.3390/healthcare10030555 and  https://doi.org/10.3390/biom12040520

Author Response

(The authors gave the same response as above.)
